# The Management of Inflammatory Bowel Disease during Reproductive Years: An Updated Narrative Review

**Nariman Hossein-Javaheri [1], Michael Youssef [2], Yaanu Jeyakumar [2], Vivian Huang [3,†] and Parul Tandon [3,*,†]**

1   Department of Internal Medicine, University at Buffalo, The State University of New York, Buffalo, NY 14203, USA; narimanh@buffalo.edu
2   Department of Internal Medicine, University of Toronto, Toronto, ON M5S 3H2, Canada
3   Division of Gastroenterology and Hepatology, Mount Sinai Hospital, University of Toronto, Toronto, ON M5G 1X5, Canada
*   Correspondence: parul.tandon@uhn.ca
†   These authors contributed equally to this work.

**Abstract:** Inflammatory bowel disease (IBD) frequently affects women of childbearing age and often coincides with pregnancy. With an increased incidence of IBD, gastroenterologists and obstetricians are more frequently involved in caring for women of reproductive age. While the development of novel therapies has allowed for successful conception and pregnancy outcomes, many patients may hesitate to conceive due to concerns for presumed adverse IBD effects on maternal and fetal health. As such, a noticeable percentage of patients may choose voluntary childlessness. Indeed, active IBD carries a greater risk of adverse pregnancy outcomes, including a loss of pregnancy, preterm delivery, and emergent C-sections. However, those with a quiescent disease tend to have fewer pregnancy complications. Therefore, it is essential to achieve remission prior to conception to optimize pregnancy outcomes. Dedicated IBD and pregnancy clinics can greatly assist in improving patient knowledge and attitudes towards pregnancy; through individualized pre-conception counseling, education, and medication adherence, the risks of poor pregnancy outcomes can be minimized. Furthermore, it is important for healthcare providers to have a sufficient understanding of the medication safety and tools to measure the disease activity, while counseling patients during gestation and breastfeeding periods. This review article aims to provide the most recent evidence-based management methods for IBD during pregnancy.

**Keywords:** IBD; pregnancy; Crohn's Disease (CD); Ulcerative Colitis (UC)

## 1. Introduction

Inflammatory bowel disease (IBD) is characterized by chronic and intermittent inflammation of the gastrointestinal (GI) tract in genetically susceptible individuals. The two major types of IBD are Crohn's Disease (CD) and Ulcerative Colitis (UC). The peak incidence of IBD is between ages 15 and 29, which overlaps with child-bearing years [1]. Patients with IBD, particularly those with active disease, are at an increased risk of adverse pregnancy outcomes, including a loss of pregnancy, infants born small for gestational age (SGA), preterm delivery, preterm pre-labor rupture of membranes (PPROM), and emergent caesarean deliveries [1–3]. Given its risks, potential harms to the child, and the need for medication use, women of child-bearing age may be hesitant to become pregnant [4]. As such, it is essential for both anticipating mothers and healthcare providers to be aware of the effects of IBD on pregnancy and its appropriate therapies before conception. This article aims to provide healthcare providers with evidence-based data on the management of IBD during pregnancy.

## 2. Knowledge and Attitude

Patients with IBD may not have adequate knowledge of the effects of their disease on pregnancy. As high as 51% of patients have a lack of understanding of the IBD-related outcomes of pregnancy [5]. More than 60% may be concerned about serious complications and 40% have some fear of infertility [6]. An estimated 31% of IBD patients do not breastfeed, as many believe medications or IBD itself can be harmful to the child [6,7]. Such beliefs may explain why at least 20% of patients stop their medication during pre-conception and pregnancy [8]. Furthermore, 17% of patients may forgo pregnancy and choose voluntary childlessness, which is significantly higher compared in these patients compared to the general population, at 6% [9]. The rates of voluntary childlessness are greater with CD (19.3%) compared to UC (13.9%), and may be directly associated with disease-related knowledge [10].

## 3. Fertility and Fecundability

Infertility refers to the inability to conceive a child within 12 months of unprotected intercourse [11]. The rate of infertility worldwide among the non-IBD population is between 8 and 12%, which is similar to those with quiescent disease and no prior pelvic surgeries [3,12]. However, non-physiological factors, such as voluntary childlessness, may negatively influence pregnancy rates and conception by 17–44% [13]. As such, psychosocial factors, including disease awareness, should be considered while discussing fertility.

In the setting of active disease, fertility can be impaired due to pelvic, ovarian, and fallopian tube inflammation, dyspareunia, and sexual dysfunction [14,15]. If unresponsive to medical management, as high as 30% of patients may undergo a proctocolectomy with ileal pouch anal anastomosis (IPAA) or end ileostomy [16]. In a recent systematic review, those who had undergone IPAA had a relative risk of infertility of 4.17 (95% CI: 1.99–8.74), which may be attributed to post-surgical adhesions [17,18]. Furthermore, end ileostomies may be complicated by uterine compression, uterine prolapse, and stomal obstruction, particularly in the third trimester [19]. If an IPAA is required before conception, a laparoscopic procedure is preferred, as it may lower the risk of infertility [19]. It is important to be aware, however, that a recent meta-analysis reported a high risk of bias in previous studies assessing the impacts of surgery and infertility and future prospective studies are required to better characterize this causal association [20].

### Assisted Reproductive Technology in IBD

Given all the potential factors leading to infertility, patients of reproductive age with IBD may benefit from early referral for Assisted Reproductive Technologies (ART) [15,21]. ART is a safe and effective approach to addressing infertility in women with UC and CD [22,23]. However, it may be more beneficial for patients who have had no history of pelvic surgeries; generally, live birth rates after ART are comparable with those with and without IBD, but they are decreased by at least 50% in those with CD and a history of pelvic surgeries [21,23,24]. Considering the increased probability of infertility in this population, it is beneficial to seek early evaluation from reproductive endocrinology providers as early as six months from the initial attempt to conceive [15].

## 4. Dedicated IBD and Pregnancy Clinics

Dedicated pregnancy and IBD-focused clinics offer significant benefits in that they may lead to improvements in fertility and improved pregnancy outcomes [25,26]. Almost a third of patients referred to these clinics are women with quiescent disease who tend to have better pregnancy outcomes [25]. Even in this group, early clinical evaluation has been associated with increased medication adherence, improved prenatal care, and decreased IBD flares during pregnancy [26–28]. Although less than 50% of patients may have access to such clinics, receiving direct care from IBD gastroenterologists, obstetricians, dieticians, and psychologists can reduce voluntary childlessness and adverse pregnancy outcomes [27,28]. Furthermore, appropriate counseling and education can greatly improve IBD-associated

knowledge and attitudes towards pregnancy [29,30]. Healthcare providers should consider appropriate referrals to specialized IBD–pregnancy clinics, particularly for those with high-risk and complex pregnancies in the setting of active inflammation [12,26,31].

## 5. Effect of IBD on Pregnancy

Studies have generally demonstrated that patients with quiescent to mild IBD have similar pregnancy outcomes to the general population [32–34]. However, patients with active IBD have significantly higher rates of adverse pregnancy outcomes. O'Toole et al. demonstrated that women with IBD had higher pooled odds of preterm delivery (OR: 1.85; 95% CI: 1.67–2.05), stillbirth (OR: 1.57; 95% CI 1.03–2.38), and a birth small for gestational age (SGA) (OR: 1.36; 95% CI: 1.16–1.60) [35]. There was a slight increased risk of congenital anomalies [OR: 1.29, 95% CI 1.05–1.58], although this may be related to publication bias [35]. A more recent meta-analysis demonstrated higher odds of gestational diabetes in women with IBD (OR 2.96; 95% CI: 1.47–5.98), regardless of corticosteroid use [1]. Further potential complications include preterm pre-labor rupture of membrane (PPROM) (OR: 12.10; 95% CI: 2.15–67.98) and increased caesarean deliveries (OR: 1.79, 95% CI, 1.16–2.77) compared to healthy controls [1]. A retrospective study also demonstrated higher ectopic pregnancies in CD but not UC, and this was independent of prior abdominal or pelvic surgeries [36]. The data on pre-eclampsia are conflicting. One cohort study demonstrated twice the risk of severe pre-eclampsia in IBD, but the overall preeclampsia rate was similar between IBD and healthy individuals [37]. In a recent meta-analysis, patients with active IBD had almost twice the risk of SGA, early pregnancy loss, preterm deliveries, and stillbirths compared to those with quiescent disease [38]. As such, it is recommend that patients with IBD, particularly those with active disease, be managed in a multi-disciplinary clinic with gastroenterologists, obstetricians, and maternal–fetal medicine specialists to reduce the risk of these adverse pregnancy outcomes [26].

## 6. Effect of Pregnancy on IBD

Pregnancy is associated with hormonal, microbial, and immunological changes, which may affect the disease activity in the intestinal tract [39]. There are also data suggesting that CD and UC may behave differently during pregnancy. A prospective European study proposed no difference in the disease course during pregnancy and post-partum between pregnant and non-pregnant patients with CD [40]. In contrast, pregnant patients with UC had higher relapses during pregnancy (RR 2.19) and postpartum (RR 6.22) than non-pregnant UC patients [40]. These relapses were observed mainly in the first and second trimesters [40]. Studies have also demonstrated that IBD activity is closely related to the preconception disease state. A meta-analysis demonstrated that patients who conceive when their disease is active are more likely to have flares during pregnancy than those who conceive while in remission [41]. This association persisted in patients with CD or UC [41]. Furthermore, a recent Danish study concluded that disease activity during pregnancy was associated with the UC phenotype and disease activity in a previous pregnancy and/or within 6 months prior to conception [42]. As such, it remains critical to achieve remission prior to conception, as recommended by the current ECCO guidelines [43]. In cases of unplanned pregnancies during active inflammation, a heightened urgency is required to establish disease control with appropriate therapeutic intervention as quickly as possible to minimize the impact on overall pregnancy outcomes [44].

## 7. Medication Use during Pregnancy

Medication adherence is essential to ensure a successful and healthy pregnancy among IBD patients [26–28]. As such, it is essential for healthcare professionals to understand the risks and benefits of a chosen therapy to ensure disease remission. A summary on the safety of medications utilized to treat IBD during pregnancy and breastfeeding is provided in Table 1.

**Table 1.** Summary of evidence for appropriate medical therapy during pregnancy and breastfeeding. A green color refers to medications that are safe and approved for use during pregnancy. Yellow refers to medications that are neither approved nor disapproved and their use must be assessed in each patient. Red refers to medications that are contraindicated in pregnancy and breastfeeding.

| Medical Treatment | Safety and Recommendations in Pregnancy | Safety and Recommendations in Breastfeeding |
|---|---|---|
| Aminosalicylates (Mesalazine, Sulfasalzine, Balsalazide, Olsalazide) | Safe <br> - Folate supplementation with sulfasalazine | Safe <br> - Discontinuation with severe neonatal bloody diarrhea |
| Thiopurines (Azathioprine, 6-Mercaptopurine) | Safe <br> - Monotherapy is recommended | Safe <br> - Neonatal anemia is possible |
| Corticosteroids | Safe <br> - Recommended only with active flares | Safe <br> - Breastfeeding 4 h after use is recommended |
| Methotrexate | Unsafe <br> - Teratogenic and associated with early pregnancy loss <br> - Contraindicated in pregnancy <br> - Discontinue 3 months before conception | Unsafe <br> - Contraindicated in breastfeeding |
| Metronidazole | Safe <br> - Recommended only in case of active flares | Unsafe <br> - Avoided due to unknown long-term effects |
| Ciprofloxacin, Amoxicillin/ Clavulanic acid | Safe <br> - Recommended only in case of active flares | Safe <br> - Recommended only in case of active flares |
| Calcineurin inhibitors (Cyclosporine, Tacrolimus) | Safe <br> - Recommended only in case of active flares | Safe |
| Anti-TNF agents (Infliximab, Adalimumab, Golimumab, Certolizumab) | Safe | Safe |
| Anti-Integrin Agents (Vedolizumab, Natalizumab) | Safe <br> - Safety data are limited | Safe <br> - Safety data are limited |
| Anti-Interleukin Agents (Ustekinumab) | Safe <br> - Safety data are limited | Safe <br> - Safety data are limited |
| Anti-Interleukin Agents (Risankizumab) | Unclear safety <br> - Safety data are limited <br> - Contraindicated in pregnancy | Unclear safety <br> - Safety data are limited <br> - Contraindicated in breastfeeding |
| Janus kinase inhibitors (Tofacitinib, Filgotinib, Upadacitinib) | Unclear safety <br> - Safety data are limited <br> - Contraindicated in pregnancy | Unclear safety <br> - Safety data are limited <br> - Contraindicated in breastfeeding |
| Sphingosine-1 phosphate receptor modulators (Ozanimod) | Unclear safety <br> - Safety data are limited <br> - Contraindicated in pregnancy | Unclear safety <br> - Safety data are limited <br> - Contraindicated in breastfeeding |

### 7.1. 5-Aminosalicylates (5-ASA)

Pregnancy: 5-ASA and sulfasalazine medications are commonly used to induce and maintain remission in IBD patients and are generally considered to be low-risk medications in pregnancy. A meta-analysis published in 2008 demonstrated no significant increase in the risk of congenital abnormalities, stillbirth, early pregnancy loss, preterm delivery, or low birth weight in pregnant patients who were exposed to 5-ASA medications [45]. Although sulfasalazine crosses the placental barrier, there is no evidence to suggest adverse fetal outcomes. Because sulfasalazine is a folic acid antagonist, supplementation with a high dose of folic acid (2 mg/day) is recommended around the time of conception to prevent fetal neural tube defects. Mesalazine is a better-tolerated medication that does not cross the placenta, and is considered to be safe during pregnancy [46]. As such, it is recommended that pregnant women with IBD on oral and/or rectal 5-ASA maintenance should maintain therapy during pregnancy. One caveat is that 5-ASA formulations containing dibutyl phthalate (DBP) should be switched to another 5-ASA drug due to the risk of fetal urogenital malformations that have been observed in animal studies [47–49].

Breastfeeding: Sulfasalazine and its metabolite sulfapyridine are secreted in the breast milk and a few cases have been reported on fetal bloody diarrhea, although this remains rare [50]. As such, 5-ASAs may be continued during breastfeeding with close monitoring of breastfed infants for allergic reactions or diarrhea [26,50].

### 7.2. Thiopurines

Pregnancy: The safety of thiopurines in pregnancy has been supported by many studies. Meta-analyses have demonstrated no increased risk of early pregnancy loss or congenital abnormalities with use of thiopurines during pregnancy [51,52]. There is an association with preterm delivery, although only a few studies have controlled for disease activity, which remained a significant confounder [51,52]. Reassuringly, recent prospective studies have concluded no increased risk of congenital malformations, early pregnancy loss, preterm delivery, low birth weight, or infections during the first year of life when exposed to thiopurines during utero [53].

Thiopurines are often combined with biologics (e.g., anti-Tumor Necrosis Factor (TNF) therapies) to optimize induction and decrease the risk of developing anti-drug antibodies to the biologic [54]. Although there have been concerns about infection with combination therapy, the recent PIANO registry demonstrated no increased pregnancy complications or infections with thiopurines, biologics, or combination therapy [53]. As such, if combination therapy is required to maintain disease remission, it can likely be continued during pregnancy to avoid relapses [55]. The discontinuation of the thiopurine and continuation of anti-TNF monotherapy may be pursued on an individualized basis, ensuring close monitoring of disease relapse during preconception and pregnancy [43].

Breastfeeding: Thiopurines are probably safe to continue using during breastfeeding. Studies have shown low or unmeasurable levels of active drug metabolites in breast milk or infant blood and no adverse events in infants exposed to thiopurines up to 6 years of follow-up [56,57]. The thiopurine peak level is reached within 4 h after intake and the maximum exposure to the infant is <1% of the total maternal dose [58,59]. As such, if truly desired, mothers may choose to breastfeed at least 4 h after drug ingestion [58]. The overall detectable concentration of thiopurines in breastmilk is minimal, but asymptomatic neutropenia in breastfed infants of mothers on azathioprine has been reported [59,60]. However, it is acceptable to continue thiopurines while breastfeeding [26].

### 7.3. Corticosteroids

Pregnancy: Corticosteroids are commonly used to induce remission when patients experience disease flares, although they should not be used long-term due to their side effects. Prior studies have demonstrated an association with cleft palates in the offspring of mothers using steroids during pregnancy [61]. However, later studies have demonstrated no increased risk of cleft palates or other major congenital anomalies (heart, limb, and

genitals) with steroid use [62,63]. Data from the PIANO study reported that corticosteroid use was associated with an increased risk of preterm delivery, SGA, low birth weight, intrauterine growth restriction, and neonatal intensive care unit (NICU) admissions [64]. However, the reported adverse effects could have been attributed to inflammation in the setting of active disease leading to subsequent complications. The use of short-duration corticosteroids to manage acute IBD flares in pregnancy is generally considered to be acceptable, but patients should be monitored for any potential side effects [43]. Alternatively, budenosine could be another option for patients with mild disease, due to its targeted action in the intestine and minimal systemic absorption [65].

Breastfeeding: The corticosteroid levels in breast milk are very low and no adverse events have been reported in the infants of mothers using steroids while breastfeeding [66]. There is a general recommendation to avoid breastfeeding for 4 h after drug intake, but such recommendations are not applicable to IBD, and patients may continue to breastfeed without limitations [26].

### 7.4. Methotrexate

Pregnancy: Methotrexate is a folic acid antagonist and is considered to be teratogenic, with studies demonstrating associations with pregnancy loss and congenital malformations, including central nervous system abnormalities [67]. The use of methotrexate is contraindicated in pregnancy and women of child-bearing age should be counselled on the appropriate use of contraception [68]. Specifically, health professionals should advise patients to wait at least 3 months after stopping methotrexate before conception and to continue the appropriate folic acid supplementation throughout the course of pregnancy [69,70].

Breastfeeding: Although methotrexate is passed in small amounts through the breast milk, most guidelines recommend against its use in women who are breastfeeding [69,71,72].

### 7.5. Antibiotics

Pregnancy: Antibiotics such as metronidazole and ciprofloxacin are indicated for treatment of infectious complications associated with IBD, including perianal abscesses, fistulas, and acute pouchitis [73]. Certain studies have suggested that metronidazole may lead to cleft deformities and preterm delivery [74,75]. According to a more recent meta-analysis, metronidazole could be associated with a slight increase in congenital hydrocephaly, however, its use is otherwise considered to be safe, with minimal adverse pregnancy outcomes [76]. Similarly, fluoroquinolones are generally avoided during pregnancy, as they may affect the cartilage and musculoskeletal development in the fetus [77]. However, there appears to be no increased risk of birth defects, still birth, preterm delivery, or SGA with fluoroquinolone use [78,79]. Despite the evidence, the use of ciprofloxacin during pregnancy is generally avoided. Penicillin-based antibiotics such as Amoxicillin/Clavulanic acid may be reasonable alternatives for treating the infectious complications of IBD [80].

Breastfeeding: Metronidazole and ciprofloxacin are detectable in breast milk [81,82]. The potential complications of using antibiotics include infantile diarrhea, candidiasis, and microbiome alterations [83,84]. However, a short-term course of antibiotic therapy while breastfeeding is not contraindicated [85].

### 7.6. Calcineurin Inhibitors

Pregnancy: Cyclosporine and tacrolimus are calcineurin inhibitors involved in the regulation of T-cells and may be effective in treating flares of UC [86]. Both medications cross the placenta and are detectable in umbilical cord blood [87]. The data on the safety of cyclosporine during pregnancy remain limited, but it may be associated with adverse effects such as maternal hypertension, pre-eclampsia, preterm delivery, and gestational diabetes [88]. Therefore, its use is recommended only for treating severe steroid-refractory UC flares [89].

Breastfeeding: Cyclosporine is detectable in breast milk, although its levels are generally low [90,91]. To date, there are no reports of severe adverse outcomes with calcineurin inhibitors while breastfeeding [92]. As a result, breastfeeding may occur in patients with IBD who are undergoing therapy with calcineurin inhibitors [93].

### 7.7. Biological Agents

Anti-TNF (Tumor Necrosis Factor) Agents

Pregnancy: Anti-TNF agents are widely used for induction and maintenance therapy in IBD, specifically in those with moderate to severe disease [94]. These agents include infliximab (IFX), adalimumab (ADA), certolizumab pegol, and golimumab. Most data investigating the safety of TNF inhibitors involve IFX and ADA [95–97]. Both agents are monoclonal IgG antibodies that cross the placenta, particularly after the second trimester [98,99]. Depending on the drug, trough concentration levels may vary throughout pregnancy; the levels of IFX may increase, but ADA levels may remain unchanged [98]. Postpartum IFX and ADA IgGs are present at high levels in infants' serum and may be detectable from 6 to 12 months following delivery [100,101]. Compared to IFX and ADA, however, certolizumab appears to have the lowest degree of placental transfer [100]. In two recent meta-analyses, the safety of anti-TNF therapy during pregnancy was confirmed. Particularly, anti-TNF exposure did not increase the risk of early pregnancy loss, preterm delivery, still birth, SGA, and congenital malformations compared to the general population [102–104]. Although the data are limited for certolizumab and golimumab, they have demonstrated similar safety profiles [105,106]. Since certolizumab is a PEGylated Fab, it does not cross the placenta and is not detectable in utero [106].

Although therapy with anti-TNF agents may result in an increased risk of maternal infection, the early discontinuation of therapy or an adjustment of the intervals between doses may lead to greater rates of disease flares [107–110]. As such, it is widely suggested to continue these therapies without interruption during pregnancy [26,27].

Breastfeeding: Anti-TNF antibodies are detectable at small quantities in breast milk, but are considered to be safe during breastfeeding [111].

### 7.8. Anti-Integrin Agents

Pregnancy: Anti-integrin agents target the transmembrane receptors on immune and endothelial cells, reducing the interactions between leukocytes and intestinal blood vessels. Primarily blocking the α-4 integrin, Natalizumab was the first drug approved for CD, but is not widely used due to the potential risk of multifocal leukoencephalopathy [112]. Meanwhile, vedolizumab is a more specific agent to the intestinal tract and has fewer systemic side-effects; therefore, it is more commonly used [112]. Vedolizumab infusion in animal models has no teratogenic effects [113]. Compared to IFX, vedolizumab crosses the placenta to a lesser degree, resulting in lower detectable levels with cord sampling [114]. The maternal clearance of vedolizumab is also increased with pregnancy and its levels are not detectable in the fetus by 15 weeks post-delivery [98]. In a recent study comparing the safety of IFX and vedolizumab, no adverse pregnancy outcomes were reported [95].

Breastfeeding: In one study by Moens et al., 12 out of 23 newborns of mothers on vedolizumab were breastfed. Although the cohort was small, no adverse reactions or complications while breastfeeding were reported [95].

### 7.9. Anti-Interleukin Agents

Pregnancy: Blocking pro-inflammatory interleukins 12 and 23 is another option in the treatment of IBD. Ustekinumab is an IgG monoclonal humanized antibody that binds to the p40 subunit of IL12/23, inhibiting downstream signaling activity [115]. The data on the safety of ustekinumab during pregnancy remain minimal. One study on pregnant macaques demonstrated no adverse maternal and/or fetal effects of ustekinumab, though the medication was detectable in infant macaques 120 days postpartum [116]. In humans, the safety data on ustekinumab are mainly limited to case reports and case series. A recent

prospective observational study reported approximately 80% live births and 20% early pregnancy loss in mothers being treated with ustekinumab [114]. A systematic review of 44 studies by Gorodensky et al. on ustekinumab exposure during pregnancy in patients with CD and psoriasis did not suggest any adverse risks [117].

Risankizumab is an IL23 inhibitor that has been approved as a maintenance therapy for moderate to severe CD [118]. The data on the safety of risankizumab during pregnancy remain minimal. In a recent study by Ferrante et al., three pregnancies in patients receiving risankizumab were reported on [119]. Two pregnancies were uncomplicated and one was voluntarily terminated due to fetal defects (cystic hygroma and hydrops fetalis) [119]. In patients with psoriasis, it is recommended not to conceive until at least 21 weeks after their last administered dose [120]. According to a recent report by Abvvie pharmaceuticals that included 60 patients on risankizumab, 11 had early pregnancy loss, 2 had elective terminations, and 18 had live births without complications [85]. Although the data on risankizumab are limited, given that the medication is a human IgG monoclonal antibody, its safety profile is expected to be similar to other human IgGs [85,121]. Currently, a definite recommendation on the continuation of risankizumab in pregnant women with IBD cannot be provided; its continuation must be evaluated on a case by case basis.

Breastfeeding: The transfer of maternal ustekinumab to the newborn through breastfeeding is a possibility. However, its concentrations in breast milk are estimated at 1/1000–1400th of the maternal serum [116,122]. Although extremely limited, to date, there are no reports on adverse effects of ustekinumab on children with breastfeeding [122–124].

*7.10. Janus Kinase Inhibitors*

Pregnancy and breastfeeding: Janus kinase (JAK) inhibitors block the action of intracellular tyrosine kinases. Tofacitinib, upadacitinib, and filgotinib have been approved for treating moderate to severe UC [125]. Although JAK inhibitors are associated with potential infections, thromboembolism, adverse cardiovascular events, and malignancy, the understanding of their safety during pregnancy is limited [126]. As a small molecule, JAK inhibitors potentially cross the placenta. Animal studies have suggested increased teratogenic effects of tofacitinib [127]. However, based on a human study on rheumatoid arthritis and psoriasis, exposure to tofacitinib during conception is not likely to be associated with increased risks to the fetus [128]. Among those with UC, a small study on 34 pregnancies demonstrated no increased risk of congenital malformations [129]. Currently, tofacitinib is not approved for use in pregnancy or breastfeeding, and the manufacturer has recommended contraception for at least 4 to 6 weeks after the last dose [130].

*7.11. Sphingosine-1 Phosphate Receptor Modulators*

Pregnancy and breastfeeding: Ozanimod is the first oral agonist of the sphingosine-1 phosphate (S1P) receptor approved for the treatment of moderate to severe active UC. Its safety in humans has not been established yet; however, during clinical trials, a single case of early pregnancy loss was attributed to ozanimod use [131]. In animal studies, binding S1P receptors interrupts the organogenesis of blood vessels and the heart due to a potential teratogenic effect [132]. Some metabolites of ozanimod have a longer half-life; therefore, its discontinuation at least three months prior to a planned pregnancy is recommended in the treatment of multiple sclerosis [133]. Similarly, in IBD, given the lack of human data, ozanimod use during pregnancy and breastfeeding is contraindicated and should be stopped at least three months prior to conception [43,85,133].

## 8. Measuring Disease Activity during Pregnancy
### 8.1. Labwork

Monitoring for IBD disease activity during pregnancy requires clinicians to consider changes in normal values during pregnancy. For example, a recent systematic review demonstrated that serum albumin and hemoglobin did not correlate well with disease activity in pregnant patients with IBD [134]. While C-reactive protein (CRP) may be a

useful tool for assessing active disease in the early trimesters, it may not accurately reflect disease activity in the later trimesters [135]. Pregnancy also causes a physiologic increase in erythrocyte sedimentation rate (ESR) due to an increased fibrinogen level, limiting its use for the evaluation of disease activity in pregnancy [136]

Faecal calprotectin (FCP) is a reliable marker of gastrointestinal mucosal inflammatory activity, which can be detected prior to systemic signs of infection with a good correlation with endoscopic inflammation in IBD [137]. Several studies have demonstrated that FCP is minimally affected by pregnancy physiology and may be a reliable marker of active disease or imminent disease flare in all gestational ages of pregnancy when compared with non-invasive disease scores, including physician global assessment scores [137–139]. However, conflicting data exist, such as a prospective cohort study by Shitrit et al., which concluded that there was a poor correlation between FCP and clinical disease relapse [140,141]. Given the lack of consensus on the utility of biomarkers, clinical judgment and the use of multiple modalities, including imaging, is important for effective therapeutic decision making.

## 8.2. Imaging

Gastrointestinal ultrasonography (GIUS) should be considered for the assessment of luminal disease activity in IBD and suspected extra-luminal complications and strictures during pregnancy [142]. A multicenter observational study demonstrated that GIUS provides adequate views of the colon and terminal ileum for up to 20 weeks during gestation and offers a specificity of 83%, sensitivity of 74%, and negative predictive value of 90% compared to FCP [143].

Non-gadolinium magnetic resonance imaging (MRI) can be considered in situations where ultrasonography is inadequate. Although there are theoretical concerns with first-trimester exposure to MRI, including tissue heating and fetal hearing impairment, no such adverse events have ever been reported [144]. The safety of gadolinium-enhanced MRI during pregnancy is controversial, however. For example, a recent population-based cohort study demonstrated an increased association between gadolinium-exposed infants and rheumatological, inflammatory, or infiltrative skin conditions and stillbirth or neonatal death [145], so its use is not recommended in pregnancy.

Ionizing radiation used in computed tomography (CT) carries a potential increased risk of congenital malformation and neurodevelopmental abnormalities [146]. Although these outcomes have never been reported in both animal and human studies [146–148], the consensus is that contrast should be avoided during pregnancy [142]. However, if imaging is medically justified for a potential change in management, then its utility should be discussed on individual basis.

## 8.3. Endoscopy

The current consensus guidelines recommend that endoscopy should not be delayed during pregnancy in patients with appropriate indications, but should preferentially be performed in the second trimester due to potential safety concerns, including maternal and fetal hypoxia, aspiration, hypotension, and an increased risk of venous thrombosis [26,34,149]. Recent data suggest that endoscopy is generally low risk during pregnancy, including a systematic review by De Lima et al., which concluded that lower gastrointestinal endoscopy was of a low risk for both the mother and infant in all three trimesters, including no increased risk of adverse pregnancy events such as early pregnancy loss, preterm delivery, and fetal demise [149]. In approximately 80% of cases, lower gastrointestinal endoscopy or sigmoidoscopy can change the medical management in IBD patients when appropriately indicated [150]. Special endoscopy considerations during pregnancy include placing the mother in the left lateral position to avoid aortic and vena cava compression and fetal monitoring by obstetrical colleagues [151]. If medically necessary, endoscopy should be performed with appropriate anesthesiology assistance and obstetrician consultation for fetal monitoring [150,152].

### 9. Gut Microbiota in IBD and Pregnancy

The microbial composition of the gut has been associated with the development of chronic diseases, including IBD [153]. The gut microbiome can fluctuate in IBD, but the overall diversity of GI bacteria is reduced in patients compared to healthy individuals [154,155]. Alteration of the microbiome is likely secondary to environmental inflammation of the gut, which is more suitable for anaerobic bacteria [156]. Concomitantly, the maternal microbiome changes throughout pregnancy. While the microbiota during the first trimester are similar to those in non-pregnant women, the third trimester is associated with increased inflammatory changes and a reduced gut diversity [157]. As such, pregnant women with IBD have demonstrated an even narrower diversity in their gut biome compared to those without IBD, which directly impacts the neonatal microbiota [158]. Several sources of maternal–fetal microbial transmission have been proposed, however, the gut has the greatest influence on developing the microbial profile in infants [159].

The microbiome of the GI tract is closely associated with inflammation and immune-mediated signaling [160]. Normally, cytokine signaling is required for healthy fetal development [161]. For instance, increased TNF-$\alpha$ and decreased IL-8 levels are reported throughout pregnancy [162]. In a recent study by van der Giessen et al. on 46 pregnant IBD patients, the overall levels of pro-inflammatory markers (IL-6, IL-8, IL-12, IL17, and TNF-$\alpha$) following conception were decreased; throughout pregnancy, however, the IL-10 and IL-5 levels were reduced, but the IL-8 and interferon-gamma (IFN$\gamma$) concentrations were increased [163]. Van der Giessen et al. concluded that, with an improvement in the immunological parameters in pregnancy, the microbial diversity becomes more similar in pregnant women without IBD [163].

Appropriate dietary agents might be helpful for normalizing the gut flora and maintaining IBD remission [164]. For instance, defined and polymeric formulas rich in proteins have restored healthy microbiota and treated symptoms of intestinal inflammation in CD among pediatric patients [165,166]. Currently, the MELODY (Modulating Early Life Microbiome through Dietary Intervention in Pregnancy) trial is evaluating the efficacy of diet during the third trimester of pregnancy in normalizing gut microbiome in IBD patients and their offspring [167]. Although dietary modifications alone might not sufficiently control disease symptoms, they can be utilized as an adjunctive therapy to medications if tolerated by patients.

### 10. Mode of Delivery

Generally, vaginal delivery is recommended for women with IBD, especially in those with mild or quiescent disease [168,169]. The incidence of Cesarean section (C-section) is greater in IBD and may be more common in CD compared to UC (52% vs. 48%) [2,169,170]. It appears that the mode of delivery (C-section vs. vaginal) does not affect the disease progression in CD [171,172]. However, in the setting of IPAA and active perianal disease, a C-section may be considered. With IPAA, damage to the anal sphincter and a harmful increased pouch pressure are of concern [173,174]. However, an older study on vaginal deliveries in those with IPAA suggested no change in the functional outcome of the pouch [175]. In the setting of active perianal disease, more than 60% of patients have reported the aggravation of symptoms with vaginal deliveries, which is attributed to a greater risk of sphincter injuries [173]. With previous ileostomies/colectomies, due to concerns about adhesions and bowel obstructions, performing a C-section must be evaluated on an individual basis [176]. Overall, it is generally recommended that shared decision making between the gastroenterologist, obstetrician, and patient be pursued to determine the most optimal mode of delivery in IBD.

### 11. Breastfeeding Considerations

Breastfeeding and the use of breast milk has numerous benefits for both the mother and child and is recommended at least until 6 months of age by most pediatric societies [177]. Unfortunately, most women with IBD initiate breastfeeding in the immediate post-partum

period, but cease breastfeeding early due to perceived insufficient milk production or concerns about medication transfer through breast milk [178]. While the literature supports that breastfeeding is protective for infants born to mothers with IBD through the development of innate mucosal immunity and prevents early-onset IBD [179], there have been several conflicting studies. A recent study demonstrated that there are reduced levels of IgA antibody and lactose in the breast milk of mothers with IBD, which may theoretically dampen the benefits of breast milk in these infants [180] Additionally, higher levels of inflammatory cytokines and succinate have been seen in the breast milk of IBD patients, which may predispose infants to inflammatory diseases or negatively impact their gut microbiome [180]. There remains a need for larger prospective studies to investigate whether differences in breast milk composition can predispose infants to developing infection, IBD, or other inflammatory or immunologic diseases. It should be noted that there is no increased risk of maternal disease flare with breastfeeding, and it may in fact be protective against IBD flares [181,182].

## 12. Conclusions

Active IBD remains a predictor of poor maternal and neonatal outcomes. With an increasing incidence of IBD, especially among women of child-bearing age, it is important to be aware of recent pregnancy specific evidence-based treatments and considerations. Furthermore, the awareness of medication safety with pregnancy is essential in avoiding unwanted disease flares and medication non-adherence. Overall, ensuring disease remission prior to and throughout conception is the most ideal approach to avoiding adverse pregnancy outcomes in women living with IBD.

**Author Contributions:** Conceptualization, N.H.-J., M.Y., Y.J., P.T. and V.H.; writing—original draft preparation, N.H.-J., M.Y. and Y.J.; writing—review and edition, P.T. and V.H. All authors have read and agreed to the published version of the manuscript.

**Funding:** This research received no external funding.

**Institutional Review Board Statement:** Not applicable.

**Informed Consent Statement:** Not applicable.

**Data Availability Statement:** Not applicable.

**Conflicts of Interest:** The authors declare no conflict of interest.

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
