# Peer review of "The Management of Inflammatory Bowel Disease during Reproductive Years: An Updated Narrative Review"

_2673-3897, doi:10.3390/reprodmed4030017_

Round 1
Reviewer 1 Report
I enjoyed reading Nariman Hossein-Javaheri et al., a review article aimed to provide the most recent evidence-based management of IBD during pregnancy. As the literature is constantly expanding, they provide a comprehensive, updated overview of the literature on the management of the IBD patient during reproductive years, from conception to delivery, and provide action tips to help guide the clinicians in the management of the IBD patient before, during and after pregnancy. It is an Updated Narrative Review. I totally agree authors summarized that disease control prior to conception and throughout pregnancy are the cornerstone to successful pregnancy management in IBD patients. Although the woman with IBD possesses a greater potential for a complicated pregnancy, the majority of these patients will experience an uneventful normal pregnancy. It is important to educate the young patient with IBD during family planning counseling. Conception at a time when IBD is quiescent offers the greatest likelihood of an uncomplicated pregnancy. Physicians must recognize and inform their patients that most medications that are necessary to suppress the disease should be continued throughout pregnancy. Although generalities can be made regarding the management of pregnant women with IBD, the individual patient may need specifically tailored therapy for her individual case. Female IBD patients may have questions about their fertility especially related to the effects of their medications and disease itself. Even more concerns are raised about the pregnancy course and their baby’s safety while having this chronic intestinal condition. Preconception counseling provides an opportunity to focus on the benefits of controlling disease activity before and during conception is critically important. Involving a maternal-fetal medicine specialist early in the conversation provides more confidence for the patients as they make decisions about IBD management.
A summary of recommended IBD therapies during pregnancy and breastfeeding is well presented, in Table 1. IBD patients with refractory to pharmaceuticals inevitably may require pouch surgery (restorative proctocolectomy with ileal anal pouch anastomosis, RPC-IPAA), which may result in a higher risk of infertility.
Author Response
Dear esteemed reviewer of Reproductive Medicine,
Thank you for providing kind and positive feedback for the review article titled “The Management of Inflammatory Bowel Disease During Reproductive Years: An Updated Narrative Review”.
We, the authors, greatly appreciate your time and support.
Sincerely,
Dr. Nariman Hossein-Javaheri and Dr. Parul Tandon
Reviewer 2 Report
In this study, authors investigated the management of inflammatory bowel disease during reproductive years. Authors claimed that dedicated IBD and pregnancy clinics can greatly assist in improving patient knowledge and attitude towards pregnancy; through individualized pre-conception counseling, education, and medication adherence, risks of poor pregnancy outcomes can be minimized. They further suggest that it is important for health-care-providers to have sufficient understanding of medication safety and tools to measure disease activity while counseling patients during gestation and breastfeeding periods. Finally, this review article has aimed to provide the most recent evidence-based management of IBD during pregnancy. In general, this review paper is interesting and well designed. Here are some comments from this reviewer.
1. Microbiota dysbiosis is associated with the development of IBD, this should be involved in this review.
2. How about the immune system and immune cells in pregnancy with IBD.
Author Response
Dear esteemed reviewer of Reproductive Medicine,
Thank you kindly for providing revisions for the review article titled “The Management of Inflammatory Bowel Disease During Reproductive Years: An Updated Narrative Review”.
The following concerns have been addressed:
- Microbiota dysbiosis is associated with the development of IBD, this should be involved in this review.
- How about the immune system and immune cells in pregnancy with IBD.
- A new section titled “Gut microbiota in IBD and pregnancy” was added to the manuscript. The mentioned section is attached to this letter for your viewing. The aforementioned section has been included in the main article between lines 404-429.
We hope this added section meets your expectations. Kindly, see the attached file.
Sincerely,
Dr. Nariman Hossein-Javaheri and Dr. Parul Tandon
